# Developmental Thermal Reaction Norms of Leatherback Marine Turtles at Nesting Beaches

**DOI:** 10.3390/ani14213050

**Published:** 2024-10-22

**Authors:** Marc Girondot, Caleb J. Krueger, Camille Cléomène, Zeenat Tran, Damien Chevallier, Fredric J. Janzen

**Affiliations:** 1Laboratoire Ecologie Systématique et Evolution, Université Paris-Saclay, Centre National de la Recherche Scientifique, AgroParisTech, 91190 Gif-sur-Yvette, France; camillecleomene997@gmail.com (C.C.); zeenat.tran@universite-paris-saclay.fr (Z.T.); 2Ecology, Evolution, and Behavior Program, Department of Fisheries and Wildlife, W. K. Kellogg Biological Station, Michigan State University, 3700 East Gull Lake Drive, Hickory Corners, MI 49060, USA; kruegeca@gmail.com (C.J.K.); janzenf1@msu.edu (F.J.J.); 3BOREA Research Unit, Laboratoire de Biologie des Organismes et des Ecosystèmes Aquatiques, Muséum National d’Histoire Naturelle de Paris, Centre National de la Recherche Scientifique 8067, Sciences de l’Univers, Institut de Recherche Pour le Développement 207, Université de Caen Normandie, Université des Antilles, Campus Martinique, BP-7207, 97275 Schoelcher Cedex, Martinique FWI, France; damien.chevallier@cnrs.fr

**Keywords:** leatherback, *Dermochelys coriacea*, embryo, sex ratio, growth, temperature-dependent sex determination, population dynamics, TSD, TSD pattern, pivotal temperature, transitional range of temperatures

## Abstract

Temperature-dependent sex determination in marine turtles has garnered significant attention from biologists and conservationists since its discovery. The leatherback, the largest extant species of marine turtle, nests on beaches within the intertropical region. Understanding temperature-sensitive sex determination is crucial in the context of climate change. Several studies have presented conflicting data regarding variation in the thermal sensitivity of sex determination in this species. Utilizing recent statistical analysis methods, we demonstrate that leatherbacks from Malaysia in the West Pacific, Costa Rica in the East Pacific, and French Guiana in the Atlantic exhibit a similar pattern. We conclude that previously noted differences were primarily due to methodological differences rather than biological differences. Our understanding of sex determination patterns in this species allows us to propose an explanation for the rapid fluctuations in population size observed, both increasing and decreasing, at various nesting sites.

## 1. Introduction

Reptiles with temperature-dependent sex determination (TSD), where thermal conditions during a restricted period of embryonic development primarily influence gonadal differentiation, have a sex-determining mechanism, in contrast to genotypic sex determination (GSD), that almost by definition embodies variation. TSD was first discovered in the lizard *Agama agama* [1] and then in turtles and tortoises [2,3], which illustrated from the outset that TSD varied in pattern among taxa, with gonadal males being produced at high temperatures in *Agama* and low temperatures in those initially studied chelonian species. Since those early experiments, additional macro-level variation in patterns of TSD has been uncovered in other reptile taxa [4].

The TSD pattern is generally characterized by the pivotal temperature (PT; synonym being critical or threshold temperature but in recent years only the term “pivotal temperature” is used) that produces each sex in equal proportion and the range of temperatures that produces both sexes, named the transitional range of temperature, or TRT [5]. The limits of TRT are particularly difficult to determine as they depend on the number of eggs incubated: if the theoretical number of males at one temperature is 0.1 and 10 eggs are incubated, the probability to not observe any male is 0.34 but, if 100 eggs are incubated, it is only 10^−5^. These two variables vary among species [6]. Furthermore, variation has been detected within species, including geographically recognizable among-population/nesting area patterns in some cases [7], and even within populations both temporally [8] and among clutches [9].

It is arguably these latter within-species observations of variation in aspects of TSD that have implications for microevolution, population dynamics, and conservation. Most concerning is that reports of these results, which are of the magnitude to have meaningful impact on the important topics identified above, could vary for non-biological reasons, including variation in experimental design, analytical methods employed, criteria for gonadal assessment, etc. [10]. For example, the limits of the TRT are particularly difficult to determine, as they depend on the number of eggs incubated.

The analysis of TSD pattern began by using qualitative tools and shifted later to mathematical and statistical descriptions. The objective of models in the context of description of biological traits is to permit a generalization and a comparison between species or populations. Without a model, sex ratio data at different temperatures will only be “timber collection” data without any capacity to compare different situations if temperatures and number of sexed embryos are not the same. “Model” in this context refers to a set of equations and a set of parameters for these equations. Different equations have been used to model sex ratio according to incubation temperature [6,11]. The objective of these equations is to best represent the observations but also to be easily workable from a mathematical point of view. Different methods have been used to estimate the value of equation parameters, including maximum likelihood [6] and later Bayesian statistics [11]. This evolution is not just cosmetic but is profound in terms of consequences of model description. When maximum likelihood (ML) is used to obtain the parameter values, the concept of probability of observing data (*d*) when the model (*m*) is known is used: $p\left(d|m\right)$. Even if parameters are changed during the ML adjustment, at each step, $p\left(d|m\right)$ is used. The problem is that the model is not known because it is precisely this model that is searched for when the analysis is performed. We are interested rather in probability of the model (*m*) when observed data (*d*) are known, $p\left(m|d\right)$, and Bayesian statistics offer this point of view.

Another shift of philosophy of data description and analysis is the biological meaning of the relationship between phenotype and environment (here mainly the temperature). This relationship is often named a thermal reaction norm [11,12,13,14]. However, it is important to consider that it is not a true reaction norm because a reaction norm describes the pattern of phenotypic expression of a single genotype across a range of environments [15]. Until we have clones, the relationship between constant temperature and sex or pattern of growth is not a true reaction norm. Beyond the temperature effect (T), various factors can also affect the relationship between constant incubation temperature and sex ratio: genetic effect for any level (clutch, female, male, rookery, population, regional management unit (RMU)—a spatial entity of individuals of the same species that are linked by using the same area), rank of clutch within a series, nesting season, ordinal date in season, amplitude of temperatures due to incubation box, shelf in incubator, incubator identity, or investigator effects on the identification of sexual phenotype of the gonad. Until now all these factors have been ignored when TSD was analyzed. We introduce a new way to analyze TSD patterns using a Bayesian non-linear mixed model. This approach is now available using the STAN statistical language [16]. All sources of variation beyond the effect of temperature are added as nested random effects to describe the relationship between sex ratio and temperature, taking into account the influence of other effects. For simplicity, we continue to name the relationship of any character with temperature as a thermal reaction norm.

The objective of the current study was to update the description of the relationship between temperature and incubation conditions of leatherback marine turtles, *Dermochelys coriacea* (Vandelli, 1761). We choose this species because its TSD pattern has been under debate since its first description. In 1983, the pivotal temperature of French Guianan leatherback (Northwest Atlantic RMU) was estimated to be between 29 and 30 °C [17]. In 1985, the same authors using new data indicated that it was within 28.75 and 29.75 °C [18]. Later, again using more data, these authors concluded that “we show that phenotypic males and potential phenotypic females are obtained from the same clutch at 29.5 °C. Therefore, 29.5 °C is the threshold temperature for sexual differentiation of the gonads in *D. coriacea*.” [19]. It is interesting to note that this definition of threshold temperature is not the definition that was retained a few years later by the same leading authors [5]. The temperature of 29.5 °C described in [19] is a temperature within the TRT rather than the PT. Later, Binckley et al. [20] concluded that the “pivotal temperature for leatherback turtles nesting on the Pacific coast of Costa Rica at Playa Grande [East Pacific RMU, this precision was not in the original text] was 29.4 °C and not biologically different from that determined for Atlantic leatherbacks.” This conclusion was based on visual inspection of plots of data. Chevalier et al. [21] instead conclude that a significant difference between French Guiana/Suriname and Pacific Costa Rican leatherback TRT exists using an argument based on bootstrapping. However, Mrosovsky et al. [22] note: “Because the functions relating sex ratio to temperature were determined on relatively few eggs, and not by the same experimenters, it would be more appropriate to emphasize that transitional range of temperature differed between particular clutches in particular experiments.” The sex ratio of 34 hatchlings from four Malaysian leatherback nests (West Pacific RMU) incubated at fluctuating temperatures has been established [23]. Binckley and Spotila [24] compared the French Guiana and Pacific Costa Rican leatherback TSD patterns with the Malaysian one using an incubation temperature proxy based on the average temperature during the entire incubation. Their conclusion still favors the hypothesis of a common pattern but leaves open the possibility that TRTs were different [24]. Mrosovsky et al. [22] conclude that “Whether these findings stem from true regional differences or are individual differences or are methodologically derived is not yet established.” We agree with this conclusion and aim to evaluate it with modern statistical tools here.

Our focus was on incubation duration and growth rate of embryos dependent on incubation temperature and TSD. The growth rate was needed to convert fluctuating temperatures into constant-temperature equivalents (CTEs), but it also has a direct bearing on detecting differences between populations. Hatching success and hatchling performance are other parameters potentially impacted by incubation temperatures but are either not available in the literature (performance) or impossible to model due to lack of data and information (hatching success).

## 2. Materials and Methods

### 2.1. Bibliography Analysis

On 24/5/2024 we searched Google Scholar (Menlo Park, CA, USA) using the keywords (leatherback OR Dermochelys) AND (temperature-dependent sex determination) and received 2050 results. These references have been merged with the bibliographic database of one author (MG). We filtered results to only retain publications with recorded temperatures during incubation, hatching success, and sex ratio or embryo growth data.

#### 2.1.1. Northwest Atlantic RMU—French Guiana and Suriname

Relevant data from Suriname and French Guiana have been published in a set of journal articles, reports, symposium papers, and theses [17,18,19,25,26,27,28,29,30]. Based on the partial redundancy of the information in these publications, it is difficult to ascertain the exact number of eggs incubated under each condition (temperature and substrate quality) and the corresponding number of hatched eggs and males, females, and intersexes. These papers have been decrypted in two databases: databaseTSD in R package *embryogrowth* [31] and the ROSIE database [32]. A consensus database has been built with a total of 20 incubation conditions using 10 different temperatures and 142 eggs (databaseTSD 9.4 and ROSIE 1.0.3). This is consistent with Binckley and Spotila [24], who also identified 142 sexed embryos incubated at 10 temperatures for Northwest Atlantic leatherbacks.

Data for incubation duration at various temperatures or embryo sizes from this region are ambiguous. For example, Figure 14 in Renous et al. [33] relates incubation duration with incubation temperature. In this graph, the meaning of uncertainty bars is not indicated (SD, SE, x.SD?) and the number of eggs for each point of the graphs is also not indicated; only the number of clutches studied is shown. In Figure 13 of the same publication [33], the length in mm of embryos is reported in the graph but there is no indication in the publication on which length it was; it is certainly not the usual straight carapace length (SCL) or curved carapace length (CCL) measures because the measures are too large (>100 mm). Bars on the graph are also not documented (SD, SE, x.SD?) and the number of embryos for each temperature is not indicated; we only know it is from 3 to 10 embryos. Based on another paper by the first author [34], we supposed that uncertainty bars are standard deviations, but we do not know if the formula for sample (division by *n* − 1) or total population (division by *n*) was used. Data have been obtained from the graphics of these publications [33,34] using the software WebPlotDigitizer v. 4.6 [35].

The mean SCL of leatherback hatchlings is 59.1 mm (SD 2.0 mm, n = 100) in East Suriname and 59.5 mm (SD 2.0 mm, n = 360) in West Suriname [36]. For the same Northwest Atlantic RMU, SCL for hatchlings in US Virgin Islands is 59.2 mm, SD = 2.4 mm (n = 2617 hatchlings from 199 clutches) [37]. These values are very similar, and the small difference could be partially due to interobserver measurement variability rather than a real difference between subgroups.

#### 2.1.2. East Pacific RMU—Costa Rica

A dataset for leatherback hatchlings was published for the East Pacific RMU with eggs obtained at Playa Grande on the Pacific coast of Costa Rica. The same data have been published in an original paper [20] and later in a book [24]. The exact number of sexed embryos at different temperatures was not published and the sex ratio graphs were not strictly the same in these two publications. Published information was then used to reconcile inconsistencies in the data (see Appendix A for details) (Table 1).

#### 2.1.3. East Pacific RMU—Mexico

Biased sex ratios were published for 103 nests, but no information on the number of sexed embryos or the nest incubation temperatures was available [38,39] and we therefore were unable to infer the necessary values as in Section 2.2. These data cannot be used further for our purposes.

#### 2.1.4. West Pacific RMU—Malaysia

Sex was determined for 34 hatchlings from four Malaysian leatherback nests incubated at fluctuating temperature (West Pacific RMU) [23]. Binckley and Spotila [24] estimated the Malaysian leatherback TSD pattern based on these four nests using the average temperature during the entire incubation. However, it should be noted that average temperature during the entire incubation is an inadequate proxy of sex-determining temperatures for species with TSD for two reasons. First, sexual phenotype is only sensitive to incubation temperature during a window of development named the thermosensitive period of development for sex ratio (TSP) [5], which is located between stages 23 to 27 in leatherbacks based on the development table of Renous et al. [33]. Second, average temperature is the worst proxy of the CTE for sex ratios among the dozen tested [40] because it does not take into account the thermal variability of developmental rates that influences both sex ratios [41] and the position of the TSP during incubation [42]. Therefore, the TSD pattern of West Pacific leatherbacks cannot be considered known based on these previous analyses.

Only one temperature was recorded each day, but nychthemeral change was supposed to be very low because boxes were not subject to direct solar radiation [23]. Temperature time series have been obtained from Figure 1 of [23] using WebPlotDigitizer v. 4.6 [35]. The 4 monitored nests were either all masculinized (n = 11, 9, and 9 males, 0 females) or all feminized (n = 5 females, 0 males).

Mean hatchling SCL was reported to be 57.3 mm (n = 200; range: 51.0–64.8 mm) [43]. The standard deviation was not published but based on the distribution of observed maximum and minimum with n = 200 [44], it was around 2.4 mm.

### 2.2. Model of Embryo Growth

Incubation durations for given temperatures are available for the Northwest Atlantic [33] and West Pacific [23] RMUs. For Northwest Atlantic data, constant incubation temperatures were used, and incubation durations were available as mean SD number of days for a set of different clutches for each temperature. We generated time series of constant temperature for each hatchling from each Northwest Atlantic clutch (n = 110), and associated incubation durations were randomly obtained from a Gaussian distribution considering the mean SD value for each temperature. For West Pacific data, 4 time series of temperatures recorded within the incubation boxes were available as well as the number of hatched eggs for each box. For two boxes, a range of incubation durations was available and has been converted into standard deviation using $SD=range/\left(1.96\times 2\right)$. For the other two boxes, only average incubation durations were published and the average SD of the two previous boxes was used. We generated time series of temperatures from each box using their published temperatures, and incubation durations were obtained randomly for each egg from a Gaussian distribution based on the mean SD value for each box. A set of 70 time series was generated corresponding to the 70 hatched eggs in this experiment [23].

We calculated the thermal reaction norm of embryo growth for each RMU using the methodology of Morales-Mérida et al. [45]. In short, the values of the 4-parameter model of Schoolfield et al. [46] were fitted using MCMC Bayesian methodology with wide uniform priors, 1000 burn-in sampling, 10,000 iterations, and thin of 10 to limit the effect of autocorrelation within the Markov chain.

We inferred the timing of the TSP using the SCL of embryos for each of these 4 Malaysian nests to delineate stages 23 to 27 [27]. We then calculated the CTE during the TSP by accounting for the average of temperatures within the TSP weighted by growth rate (model 6 of [40]).

### 2.3. Pattern of TSD

The patterns of TSD were fitted using a sigmoid model. We implemented the potentially asymmetrical model named *flexit** from the *embryogrowth* package [31], a simplified version of the *flexit* model [11]. The original *flexit* model used two parameters (*K*_1_ and *K*_2_) to model asymmetry around the pivotal temperature (*P* parameter). Celsius measurement follows an interval but not a ratio system, and it follows a relative not an absolute scale. Therefore, an object at 20 °C does not have twice the energy as when it is 10 °C. Thus, degrees Celsius is a useful interval measurement but does not possess the characteristics of ratio measures like weight or distance. Consequently, there is no reason to force the PT to be at the center of the TRT in a symmetrical TSD pattern contrary to [13]. The *S*, *K*_1_, and *K*_2_ are transformed into *S*_1_ and *S*_2_ parameters that are used for the part of the curve below (*S*_1_) and above (*S*_2_) *P*. We find it more convenient to model the asymmetry directly using *S*_1_ = *S* and *S*_2_ = *S* + ∆*S*, and the transition below and above P is managed using a quasi-binary threshold function defined with $QBT={\left(1+e^{100\left(P-T\right)}\right)}^{-1}$. QBT is 0 when *P* < *T* and 1 when *P* > *T*. QBT is 0.5 when P=T. Sex ratio at a given temperature is then:$$sr\left(T\right)={\left(1+e^{\left(4\left(S+∆S.QBT\right)\left(P-T\right)\right)}\right)}^{-1}$$

P is the pivotal temperature, while S and S+∆S control the lower and upper asymptotes, respectively. The limits of the TRT are:

${LimitTRT}_{1}=P-\frac{1}{4S}ln\left(\frac{1-l}{l}\right)$ and ${LimitTRT}_{2}=P-\frac{1}{4\left(S+∆S\right)}ln\left(\frac{l}{1-l}\right)$, where *l* is the limit of sex ratios to define TRT from *l* to 1 − *l*, and the TRT width is $\left|{LimitTRT}_{1}-{LimitTRT}_{2}\right|.$

The 3 parameters of the *flexit** model have been fitted using a Bayesian MCMC model with a binomial likelihood [11].

This model is implemented in the *embryogrowth* R package (version > 9.3) and has been scripted in STAN to include random factors. Nested random factors were RMU [47], bibliographic identifier (ID), and Clutch when available (this information is not available for some of these data). The effect of random factors was applied in the slope of the 3 parameters (*S*, ∆*S*, and *P*). The bibliographic identifier is used to consider the influence of incubation condition (substrate or temperature stability) but also how the experimenters identify sexual phenotype and interpret intersexes, for example. The tested random effects were then: 1, (1|RMU), (1|RMU/ID), and (1|RMU/ID/Clutch).

### 2.4. Comparison of Models

When several models were analyzed using a single dataset, Akaike information criterion [48] corrected for small sample sizes (AICc) [49] was used to identify the best model to limit the risk of overparametrization. We then calculated weights based on AICc [49] to estimate the probability that each model is the best among those tested. AICc and Akaike weights were estimated as predictive posterior means using the thinned Bayesian MCMC distribution.

The R code of these analyses is available in GitHub: https://github.com/Marc-Girondot/Dc_Incubations (accessed on 10 October 2024).

## 3. Results

### 3.1. Bibliography Analysis

Our literature search returned 2050 documents but only 43 were retained for including incubation temperature and hatching success, sex ratio, or embryo growth data. We did not retain publications in which sex was inferred from incubation temperatures because of the circularity of reasoning and lack of a validated model to convert series of temperatures into sex ratios for leatherbacks [50].

### 3.2. Thermal Reaction Norm of Embryo Growth

Incubation durations are known for several constant temperatures in French Guiana and Suriname leatherbacks (Northwest Atlantic RMU) and for fluctuating temperatures in Malaysian leatherbacks (West Pacific RMU). Thermal reaction norms were fitted for these two datasets separately (four parameters for each) and in a single analysis (four parameters). The posterior predictive of Akaike weight is 0.535 (SE 0.009) for a single common model of thermal reaction norm of growth for both locations. The posterior thermal reaction norms for both locations are indeed very similar (Figure 1).

Embryo growth for the four Malaysian (West Pacific RMU) nests, modeled considering the fitted thermal reaction norm of growth (Figure 1), is shown in Figure 2 along with the estimated TSP. Note that since fluctuating temperatures result in variable growth rates throughout incubation, the TSP is not the middle third of incubation [42] as is still wrongly commonly said in the literature [51,52].

The posterior predictive CTEs within the TSP for the 4 Malaysian nests (Figure 2) are shown in Table 2. Note that the 95% credible intervals are very narrow, and thus medians are an appropriate summary statistic.

### 3.3. Pattern of TSD

We evaluated the presence of RMU, ID, and Clutch effects using Akaike weights based on posterior AICc. The model with an RMU random effect (1|RMU) is favored compared to the model without (1; 0.57 vs. 0.43). Additionally, the model with a Clutch random effect (1|RMU/ID/Clutch) is favored over the model without (1|RMU/ID; 0.62 vs. 0.38). When a global comparison of the four tested models is performed, the model with all random effects (1|RMU/ID/Clutch) receives the most weight (0.37). In this case, the Clutch effect is higher than the RMU and ID effects. It should be noted that RMU and ID effects are nearly impossible to separate with this dataset as only one publication relates sex ratio for West Pacific, one for East Pacific, and two for Northwest Atlantic. On the other hand, 10 categories for Clutch effect are available.

Leatherback patterns of TSD without the random effects or with the full random effects (1|RMU/ID/Clutch) are shown in Figure 3. The pivotal temperature is estimated to be 29.49 °C (SE 0.14) and *S*_1_ and *S*_2_ are very similar (2.34 °C SE 1.67 and 2.83 °C SE 1.75), indicating that the fitted pattern is very close to being symmetrical. The main pattern is obviously unchanged whether random effects are included or not, but the uncertainty levels for TRT limits as well as the pivotal temperature are higher when random effects are included. Not including these random effects increases the risk of identifying a false positive difference among sites when the difference is more appropriately attributed to a Clutch effect.

## 4. Discussion

The initial objective of this work was to describe the pattern of TSD of leatherback turtles using modern statistics. We faced substantial difficulties interpreting previously published data (see Appendix A). We concur with Janzen and Paukstis [10] and Morales Mérida et al. [53] that data presentation is of prime importance to be able to reuse TSD data. Beyond these challenges, we have analyzed the leatherback pattern of temperature-dependent growth rate from two regions (Malaysia in West Pacific and Suriname and French Guiana in Northwest Atlantic RMUs) and TSD patterns from three regions (Malaysia in West Pacific, Suriname and French Guiana in Northwest Atlantic, and Costa Rica in East Pacific RMUs).

The relationships of growth rate to temperature (West Pacific and Northwest Atlantic RMUs) are very similar and monotonically ascending according to incubation temperature (Figure 1). Although one curve was obtained using constant incubation data and the other was obtained using fluctuating temperatures, the results are very similar, confirming that the model used to estimate the instantaneous growth rate successfully captures the effect of temperature changes [54]. Interestingly, embryonic growth rate only increases with temperature (Figure 1) as it does in the olive ridley marine turtle [45]. No sublethal effect can be seen (i.e., growth rate decreasing at the warmest temperatures) in these tropical species. However, such a sublethal effect has been observed in the loggerhead, a species that lays its eggs on beaches in more temperate regions [54]. No difference in thermal reaction norm of embryo growth rate according to RMU origin has been detected for leatherback turtles.

Based on available data, we resolved the controversy about the presence or absence of differences for TSD patterns among populations (RMUs) of leatherback turtles. We conclude that TSD pattern differences exist but are more linked to study effect (incubator regulation, investigator effect about sex identification of the gonad, etc.) and clutch effect (maternal or paternal effect, egg laying period of the year, etc.) than to origin of eggs at the scale of RMUs. The conclusion about similarity of thermal reaction norm of embryo growth rate (previous paragraph) holds also for thermal reaction norm of TSD.

The leatherback TSD pattern that we calculated using mixed-model Bayesian statistics (Figure 3) appears to be different from what was estimated using more traditional statistics, but the results are not incompatible. Indeed, using RMU, study, and clutch sources of uncertainty, the limits of the TRT and PT are much wider (Figure 3) than previously thought [21]. Thus, the apparent on/off leatherback TSD pattern conforms more to the current understanding that TSD in nature experiences multiple influences within and among nests and eggs that permit production of males and females in various conditions. However, the leatherback pattern has the thinnest TRT among marine turtles [6]. It can be anticipated that the thinner the TRT, the more sensitive to temperature change is sex determination [6], with higher risk to produce a strongly biased sex ratio, but the quantification of this effect is still missing. Indeed, the conversion of time series of natural nest temperature changes into sex ratios is a complex process [50]. For example, temperature fluctuations during incubation change the position of the developmental window during which sex is sensitive to temperature (i.e., the TSP [5]) because the rate of development depends on incubation temperature. Furthermore, the sexualizing effect of temperature within the TSP is stage dependent, and the effect of temperature change on sexualization is not linear [50].

The quantitative description of the leatherback TSD pattern opens new perspectives in terms of conservation biology. This marine turtle species has a wide distribution to polar circles, but a tropical distribution for nesting sites [47,55]. Occasionally, this species rapidly changes density on nesting sites, which is largely not understood. For example, East Pacific leatherbacks collapsed at Las Baulas National Park, Costa Rica [56,57] while during the same period nesting numbers were stable in Playa de Tierra Colorada, Guerrero, Mexico, or increasing in Playa Barra de la Cruz and Playa Grande, Oaxaca, Mexico [58]. All these beaches belong to the East Pacific RMU [47]. Leatherback nesting in Malaysia (West Pacific RMU) has essentially disappeared, declining from about ten thousand nests in 1953 to only one or two nests per year since 2003 [43]. At the border of Suriname and French Guiana (Northwest Atlantic RMU), leatherbacks were probably rare in the 18th century but represented up to 50% of nesting females globally in the years 1988–1992, with up to 60,000 nests deposited annually on four beaches for a total coastline of 10 km [58]. Recently, leatherbacks have nearly disappeared from this zone while green turtles (*Chelonia mydas*) were stable in the same beaches [59]. The aggregation of leatherbacks nesting at the east of French Guiana, 250 km away from the border of Suriname, was absent in the 1980s but appeared during the 1990s [60]. It does not show a similar crash as observed west of French Guiana [59].

Thus, it seems that local phenomena at the scale of the nesting beach may explain the rapid change in the number of nesters in some regions. This implies a potential role for incubation conditions rather than more global at-sea impacts. Indeed, leatherbacks exhibit natal homing [61] and nesting philopatry [62], so a change occurring at a nesting site will have a local consequence many years later when individuals reach sexual maturity. Much of the work about the relationship between temperature and population dynamics has concentrated on the impact of TSD [63]. However, temperature also acts on hatching success, and the interplay between TSD and hatching success could be very important for population dynamics [64].

Leatherback hatching success is low compared to other species of marine turtles [65]. Many factors influence hatching success in leatherbacks such as temperature [66], mother identity [67], pollution [68], destruction of habitat by erosion [59], and unknown seasonal factors [69], and it varies from place to place. Hatching success data for leatherback eggs incubated at constant temperatures are particularly sparse, and, for Suriname and French Guiana, poor hatching success has more to do with transporting eggs from the nesting beach to distant facilities in Paris than incubation conditions at the destination (MG, pers. obs.). Much work remains to be performed to understand the population-wide consequences of the impact of temperature on both sex determination and incubation success.

For a long time, the dogma for the life history of turtles was that subadult and adult survivorship were the main drivers of population dynamics [70]. It was the logical consequence of the high survivorship of adults and the high longevity typical of species with a K-strategy. What happened with nests and offspring on the beach was thought to have minimal importance for population dynamics even if it was not negligible [71]. However, marine turtles and many freshwater chelonians as well [72] do not conform to a typical K-strategy because each female can lay hundreds, if not thousands, of eggs during her reproductive life. This reproductive strategy is more common for an r-strategy species. If the eggs have independent history, then each egg counts for very little in the population dynamics. On the other hand, if spatial or temporal covariation exists for egg survival, the correct level of analysis should not be the egg but the aggregate of eggs, which can be a nest, beach section, or total beach. Then, what is happening for incubation is no longer anecdotal for influencing population dynamics. A similar conclusion has been reached for freshwater turtles in an integrated population dynamical model; the matrix model projections undervalued the potential effect of an increase in nest survival [72]. We must change our view about population dynamics, egg incubation, and turtles to explain rapid changes occurring at various sites. It does not mean that the adult stage is not important, of course, but population dynamics must be analyzed in a more integrated context. To do so, researchers must undertake more rigorous study designs that yield the most valid data, as well as employ the most appropriate statistical analyses, for the strongest development and implementation of conservation and management practices. The egg stage is one of the most accessible life stages for turtles. Studies of sex ratio and hatching success must be conducted with the clear objectives of feeding a population dynamics model that integrates both spatial and temporal variability.

## 5. Conclusions

The description of temperature-dependent patterns in leatherbacks from three RMUs (West Pacific, Northwest Atlantic, and East Pacific) has sparked controversy regarding the existence of differences. Using Bayesian statistics, we conclude that while a difference does exist, it is more closely related to interclutch variability than to differences at the RMU scale. With this pattern in mind, we discuss the origin of the rapid changes observed at nesting sites for this species. We conclude that the population dynamics of marine turtles, as well as freshwater turtles, should be studied using a more integrative approach than is typically employed. A more systematic analysis of the relationship between the stability of the number of nests deposited by the different marine turtles and the shape of the sex ratio thermal reaction norm would allow tests of whether the hypotheses presented here can be validated.

## Figures and Tables

**Figure 1 animals-14-03050-f001:**
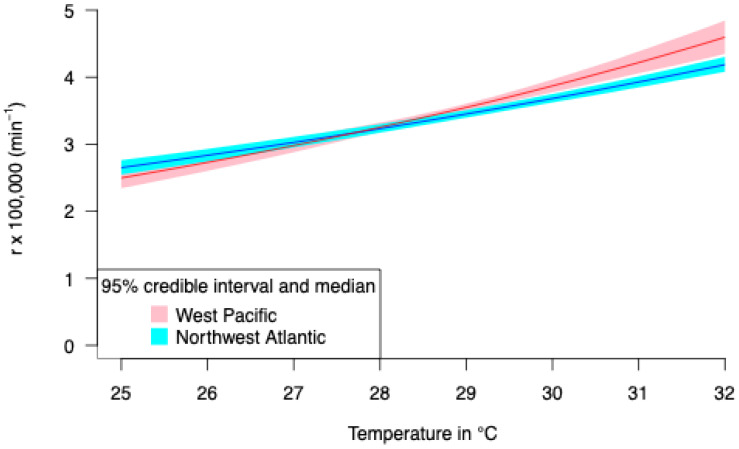
Thermal reaction norm of embryonic growth for Malaysian (West Pacific RMU) and French Guiana and Suriname (Northwest Atlantic RMU) leatherback embryos. *r* is the instantaneous growth rate in min^−1^.

**Figure 2 animals-14-03050-f002:**
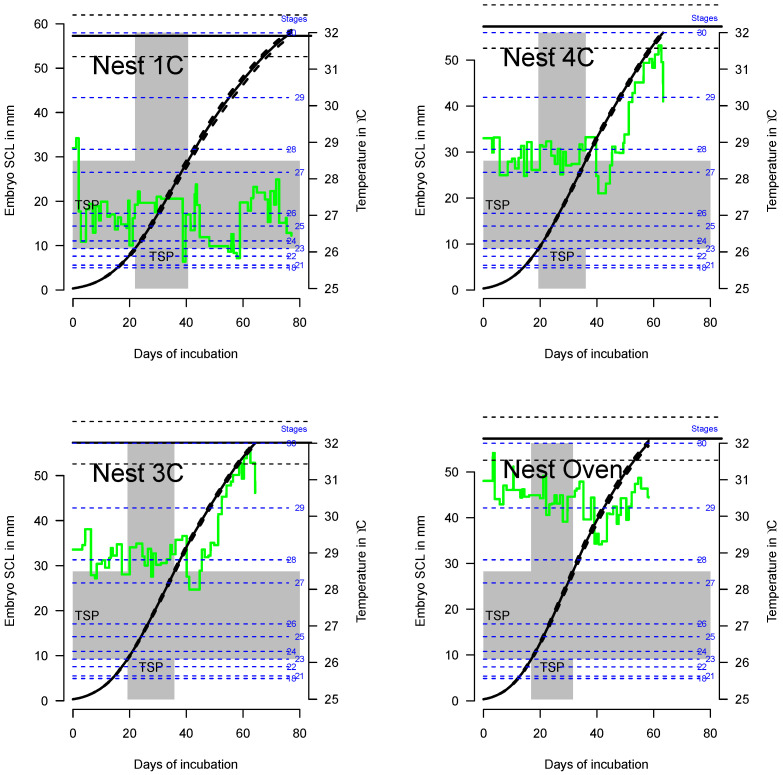
Model of embryo growth (plain black lines and 95% credible interval in dashed lines) for the 4 Malaysian nests (West Pacific RMU) with known sex ratio. The position of embryological stages as well as the TSP are shown. Note that the 4 nests have the same x- and y-scales to make comparisons easier. Green lines are the recorded temperatures, and blue lines are the embryological stages.

**Figure 3 animals-14-03050-f003:**
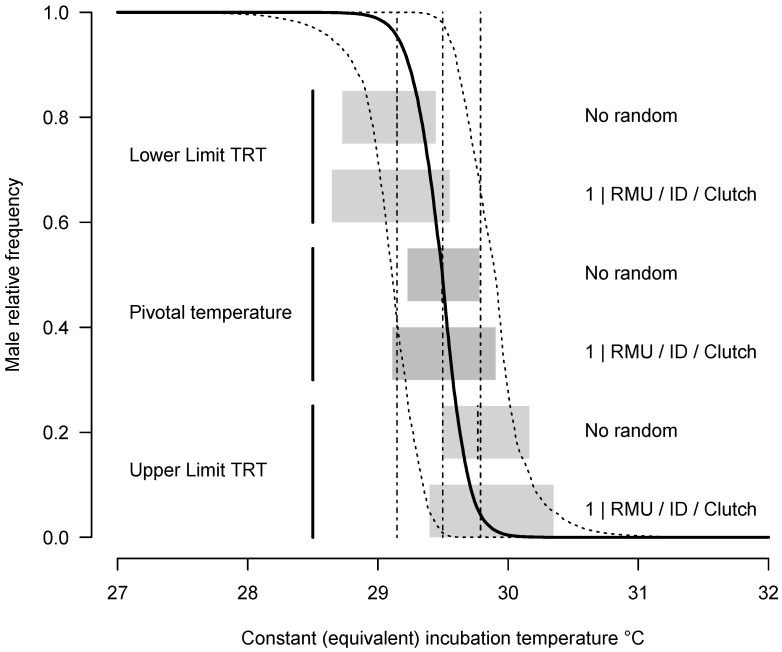
Posterior TSD pattern of leatherbacks from 3 RMUs (West Pacific, Northwest Atlantic, East Pacific). The median limits of TRT and pivotal temperature are shown in dashed lines: the lines from 0 to 1 are the medians with random effect and the lines within the boxes are the medians without random effect. Their uncertainties are shown without random effect (top gray boxes) or with random effects (bottom gray boxes). The 95% credible intervals of the pattern using 1 | RMU/ID/Clutch random effects are shown in dotted lines.

**Table 1 animals-14-03050-t001:** Interpretation of the number of sexed juveniles and males and females in databases databaseTSD [31] and ROSIE [32]. The rows in gray are the rows with uncertainty (see Appendix A).

Incubation Temperatures	Males	Females	Sexed
28 °C	17	0	17
29 °C	9	1	10
29.5 °C	4	6	10
30 °C	1	15	16
30.5 °C	0	10	10
31 °C	0	12	12
31.5 °C	0	8	8
32 °C	0	19	19
33 °C	0	0	0

**Table 2 animals-14-03050-t002:** CTE within the TSP (CTE|TSP) of the 4 Malaysian nests with known sex ratio from Chan and Liew [23]. The 95% CI is the range of posteriors corresponding to 95% credible interval.

Nest	Mean ± SE	Median CTE|TSP	95% CI CTE|TSP	Males	Females
1C	27.04 ± 0.58	27.31 °C	27.30; 27.31	11	0
3C	29.21 ± 1.01	28.90 °C	28.89; 28.91	9	0
4C	28.95 ± 1.34	28.63 °C	28.62; 28.64	9	0
Oven	30.42 ± 0.80	30.43 °C	30.43; 30.43	0	5

## Data Availability

The R code of these analyses is available on GitHub: https://github.com/Marc-Girondot/Dc_Incubations (accessed on 10 October 2024).

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
