# Peer review of "Developmental Thermal Reaction Norms of Leatherback Marine Turtles at Nesting Beaches"

_animals, 2024, doi:10.3390/ani14213050_

Round 1

Reviewer 1 Report

Comments and Suggestions for Authors

 Developmental Thermal Reaction Norms of Leatherback Marine Turtles at Nesting Beaches by Girondot et al. 2024.

The manuscript describes the temperature dependent pattern of sea turtles in general and investigates the case for Leatherback sea turtles for different populations. This study updates the understanding of temperature-34 dependent sex determination (TSD) in this species using Bayesian statistics. The differences in TSD was attributed to more likely due to study methodologies and clutch-specific factors rather than regional differences

There are some general remarks and specific comments below which can be considered in revision of the manuscript.

There are many different reports for all sea turtle species regarding the threshold temperature and corresponding sex ratios. Therefore, I suggest authors to make a comparison of such data and make general evaluation this approach for their usage for Leatherback turtles. (paragraph Line starting from L 389).

1-     Marine or sea turtle; one of them should be preferred in all parts of the Manuscript.

2. Line 69-70 the sentences need rewritten; the meaning is not that clear.

3. Line 141. The references should be given as Mrosovsky et al [25] not Mrosovsky, Kamel, Rees and Margaritoulis [25]

4. Line 374-375 the sentences needs revision, the meaning is unclear.

5. The leatherback sea turtles has the thinnest TRT among marine turtles, the thinner the TRT, the more sensitive to temperature change is sex determination, I therefore suggest the comparison of the nesting site distribution of leatherback turtles and the other turtles nesting on the same beaches. The conclusion can be supported with such information. A map of leatherbacks and presence of other sea turtles nesting on the same beach and their population levels and change over time could be useful.

6. I also suggest the comparison of egg sizes of species and the amount of air between the eggs and the depth of nest can also be considered in differences of sex ratio and TRT for leatherback turtles.

7. The conclusion can also be extended with the above suggested ideas to be included for future consideration.

Comments on the Quality of English Language

There are some remarks mentioned above should be improved.

Author Response

Referee 1

Developmental Thermal Reaction Norms of Leatherback Marine Turtles at Nesting Beaches by Girondot et al. 2024.

The manuscript describes the temperature dependent pattern of sea turtles in general and investigates the case for Leatherback sea turtles for different populations. This study updates the understanding of temperature-dependent sex determination (TSD) in this species using Bayesian statistics. The differences in TSD was attributed to more likely due to study methodologies and clutch-specific factors rather than regional differences

There are some general remarks and specific comments below which can be considered in revision of the manuscript.

There are many different reports for all sea turtle species regarding the threshold temperature and corresponding sex ratios. Therefore, I suggest authors to make a comparison of such data and make general evaluation this approach for their usage for Leatherback turtles. (paragraph Line starting from L 389).

We agree with the referee that the threshold temperatures are known for many species, for example here:

Hulin, V., Delmas, V., Girondot, M., Godfrey, M.H., Guillon, J.-M., 2009. Temperature-dependent sex determination and global change: Are some species at greater risk? Oecologia 160, 493-506.

However, it is not at all the same subject as the one described in our manuscript. Furthermore, the referee proposed to compare the threshold estimates with the corresponding sex ratios. If these sex ratios are observed using constant incubation temperature, the description is simply the thermal reaction of sex ratio. However, if these sex ratios come from nature, it is much more complicated because there are very few sex ratio data of good quality. Most of the sex ratio data in natural condition come from crude conversion of incubation temperatures into sex ratio with high level of errors. See for example:

Fuentes, M.M.P.B., Monsinjon, J., Lopez, M., Lara, P., Santos, A., dei Marcovaldi, M.A.G., Girondot, M., 2017. Sex ratio estimates for species with temperature-dependent sex determination differ according to the proxy used. Ecological Modelling 365, 55-67.

Monsinjon, J., Guillon, J.-M., Wyneken, J., Girondot, M., 2022. Thermal reaction norm for sexualization: the missing link between temperature and sex ratio for temperature-dependent sex determination. Ecological Modelling 473, 1-7.

No change in the manuscript.

  • Marine or sea turtle; one of them should be preferred in all parts of the Manuscript.

The occurrence of “sea turtles” line 17 have been replaced by “marine turtles”.

  1. Line 69-70 the sentences need rewritten; the meaning is not that clear.

The sentence has been changed to:

It is arguably these latter within-species observations of variation in aspects of TSD that have implications for microevolution, population dynamics, and conservation.

  1. Line 141. The references should be given as Mrosovsky et al [25] not Mrosovsky, Kamel, Rees and Margaritoulis [25]

Change has been done. Same problem has been identified Line 212:

Renous, Rimblot-Baly, Fretey and Pieau has been changed to Renous et al. (See answer to referee 3).

  1. Line 374-375 the sentences needs revision, the meaning is unclear.

Sentences have been changed:

Interestingly, embryonic growth rate only increases with temperature (Figure 1) as it does in the olive ridley marine turtle [48]. No sublethal effect can be seen (i.e. growth rate decreasing at the warmest temperatures) in these tropical species. However, such a sublethal effect has been observed in the loggerhead, a species that lays its eggs on beaches in more temperate regions [58]. No difference of thermal reaction norm of embryo growth rate according to RMU origin has been detected for leatherback turtles.

  1. The leatherback sea turtles has the thinnest TRT among marine turtles, the thinner the TRT, the more sensitive to temperature change is sex determination, I therefore suggest the comparison of the nesting site distribution of leatherback turtles and the other turtles nesting on the same beaches. The conclusion can be supported with such information. A map of leatherbacks and presence of other sea turtles nesting on the same beach and their population levels and change over time could be useful.

We completely agree to this proposition. However, by itself it is a full research program. We open this possibility into the conclusion.

A more systematic analysis of the relationship between the stability of beach use by the different marine turtles and the shape of the sex ratio thermal reaction norm would allow to test whether the hypotheses presented here can be validated.

  1. I also suggest the comparison of egg sizes of species and the amount of air between the eggs and the depth of nest can also be considered in differences of sex ratio and TRT for leatherback turtles.

We agree with the referee that these points should be taken into account when studying the evolution of the thermal reaction norm of the sex ratio. However, such a study cannot be done with only one species because it makes sense only when several species are analysed conjointly.

We'd like to point out, however, that sea turtle eggshells are flexible and there's virtually no air between them in a natural nest.

No change in the manuscript.

  1. The conclusion can also be extended with the above suggested ideas to be included for future consideration.

We have added a sentence in conclusions.

A more systematic analysis of the relationship between the stability of beach use by the different marine turtles and the shape of the sex ratio thermal reaction norm would allow to test whether the hypotheses presented here can be validated.

Reviewer 2 Report

Comments and Suggestions for Authors

The manuscript entitled "Developmental Thermal Reaction Norms of Leatherback Marine Turtles at Nesting Beaches" clearly describes a collection of data that leads to valuable conclusions about leatherback TSD. Notably, while the authors face significant challenges in addressing missing or incomplete data, they address these challenges through mathematical models that allow for a better understanding of how the data could be grouped and what they could be used for.

Line 79. Does this refer to nests in situ or ex situ? Please explain.
Line 103. Other factors, such as beach type and protection status, were not considered. Additionally, the beach's sand composition was not studied or proposed. Explain.
Line 106. In situ or ex situ?
Line 183. How do you view values and their application?

Line 194. What does this mean?

Line 215. If the data is variable and inconsistent, there's no point in including it in the analysis. Explain

Line 234. Please provide a more detailed description of the procedure.

Line 295. Please explain the specific location on the beach where these temperatures were recorded and whether the positions were standardized for the temperature measurements.

Line 363. Consider the impact of latitude on the leatherback compared to the other species and discuss its influence.

Line 441. A guideline of parameters should be established to allow authors to make a more accurate evaluation by providing a comprehensive set of criteria for assessment. What do you think? Explain.

Author Response

Referee 2

The manuscript entitled "Developmental Thermal Reaction Norms of Leatherback Marine Turtles at Nesting Beaches" clearly describes a collection of data that leads to valuable conclusions about leatherback TSD. Notably, while the authors face significant challenges in addressing missing or incomplete data, they address these challenges through mathematical models that allow for a better understanding of how the data could be grouped and what they could be used for.

Line 79. Does this refer to nests in situ or ex situ? Please explain.

Line 79 was: Without a model, data on sex ratio at different temperatures will be only “timber collection” data without any capacity to compare different situations if temperatures and number of sexed embryos are not the same.

It refers to ex situ data. This precision is added

Without a model, sex-ratio data at different constant temperatures will only be “timber collection” data without any capacity to compare different situations if temperatures and number of sexed embryos are not the same.

Line 103. Other factors, such as beach type and protection status, were not considered. Additionally, the beach's sand composition was not studied or proposed. Explain.

These factors are not relevant for determination of thermal reaction norm of sex ratio because thermal reaction norm is an internal factor of the individuals. The factors “beach type”, or “sand color” are important when in situ sex ratios are used. This is not the case here.

Line 106. In situ or ex situ?

Thermal reaction norm is defined in this context only for constant incubation temperatures, so it is ex situ.

This precision is added:

Until we have clones, the relationship between constant temperature and sex or pattern of growth is not a true reaction norm.

Line 183. How do you view values and their application?

This precision has been added:

Data have been obtained from the graphics of these publications [36, 37] using the software WebPlotDigitizer [38].

Line 194. What does this mean?

The sentence was:

We reconciled inconsistencies in the data when possible (see appendix A for details) (Table 1).

We changed it to (based on the recommendation of the third referee):

The exact number of sexed embryos at different temperatures was not published and the sex ratio graphs were not strictly the same in these two publications. Published information was then used to reconcile inconsistencies in the data (see appendix A for details) (Table 1).

Line 215. If the data is variable and inconsistent, there's no point in including it in the analysis. Explain

When we wrote “Therefore, the TSD pattern of West Pacific leatherbacks should be considered unknown at this stage.”, we referred to the previously published analyses. To make it clearer, we change this sentence to:

Therefore, the TSD pattern of West Pacific leatherbacks cannot be considered known based on these previous analyses.

Line 234. Please provide a more detailed description of the procedure.

The referee asked more details of this procedure:

“For West Pacific data, 4 timeseries of temperatures recorded within the incubation boxes were available. For each box, a range of incubation durations was available as well as the number of hatched eggs. When the range of incubation durations was not published, we used the average value of the boxes with this information.”

We agree that it was not enough precise. We explain better how these data have been analyzed:

For Northwest Atlantic data, constant incubation temperatures were used, and incubation durations were available as mean-SD number of days for a set of different clutches for each temperature. We generated timeseries of constant temperature for each hatchling from each Northwest Atlantic clutch (n = 110), and associated incubation durations were randomly obtained from a Gaussian distribution considering the mean-SD value for each temperature. For West Pacific data, 4 timeseries of temperatures recorded within the incubation boxes were available as well as the number of hatched eggs for each box. For two boxes, a range of incubation durations was available and has been converted into standard deviation using . For the other two boxes, only average incubation durations were published and the average SD of the two previous boxes was used. We generated timeseries of temperatures from each box using their published temperatures, and incubation durations were obtained randomly for each egg from a Gaussian distribution based on the mean-SD value for each box. A set of 70 timeseries was generated corresponding to the 70 hatched eggs in this experiment [26].

Line 295. Please explain the specific location on the beach where these temperatures were recorded and whether the positions were standardized for the temperature measurements.

No temperature measurement was obtained for these data on the beach. To model the thermal reaction of sex ratio, only ex situ data are used.

No change in the manuscript.

Line 363. Consider the impact of latitude on the leatherback compared to the other species and discuss its influence.

The referee is right that this point is important. We change the manuscript to give more precise information about latitude influence. However, it is difficult to generalize with only 3 species.

Interestingly, the growth rate of embryo is only increasing with temperature (Figure 1) as it does in the tropical olive ridley marine turtle [48]. No sublethal effect can be seen (i.e. growth rate decreasing while temperature increased) in these species that seem particularly resistant to high temperatures. Such a sublethal effect has been observed in the loggerhead, a species that lays its eggs on beaches at higher latitudes, in more temperate regions [58]. No difference of thermal reaction norm of embryo growth rate according to RMU origin has been detected for leatherback turtles.

Line 441. A guideline of parameters should be established to allow authors to make a more accurate evaluation by providing a comprehensive set of criteria for assessment. What do you think? Explain.

We add this precision:

Eggs stage is one of the most accessible life stages for turtles. Studies of sex ratio and hatching success must be conducted with the clear objectives of feeding a population dynamics model that integrates both spatial and temporal intra- and inter-variability.

Reviewer 3 Report

Comments and Suggestions for Authors

The manuscript "Developmental Thermal Reaction Norms of Leatherback Marine Turtles at Nesting Beaches" provides new techniques and interpretations regarding the temperature-dependent sex determination criteria in this and other marine turtle species. The manuscript is well-written and very thorough and I have only minor comments and suggestions in the attached draft.

Author Response

Referee 3:

All the propositions indicated on the text have been retained (see text).
